# Protective Cultures in Food Products: From Science to Market

**DOI:** 10.3390/foods12071541

**Published:** 2023-04-05

**Authors:** Sebastian W. Fischer, Fritz Titgemeyer

**Affiliations:** Department of Food, Nutrition and Facilities, FH Muenster, Correnstr. 25, 48149 Münster, Germany

**Keywords:** lactic acid bacteria, protective cultures, food safety, legal framework, market overview, bacteriocins, bio preservation

## Abstract

An ultimate goal in food production is to guarantee food safety and security. Fermented food products benefit from the intrinsic capabilities of the applied starter cultures as they produce organic acids and bactericidal compounds such as hydrogen peroxide that hamper most food pathogens. In addition, highly potent small peptides, bacteriocins, are being expelled to exert antibiotic effects. Based on ongoing scientific efforts, there is a growing market of food products to which protective cultures are added exclusively for food safety and for prolonged shelf life. In this regard, most genera from the order *Lactobacillales* play a prominent role. Here, we give an overview on protective cultures in food products. We summarize the mode of actions of antibacterial mechanisms. We display the strategies for the isolation and characterization of protective cultures in order to have them market-ready. A survey of the growing market reveals promising perspectives. Finally, a comprehensive chapter discusses the current legislation issues concerning protective cultures, leading to the conclusion that the application of protective cultures is superior to the usage of defined bacteriocins regarding simplicity, economic costs, and thus usage in less-developed countries. We believe that further discovery of bacteria to be implemented in food preservation will significantly contribute to customer’s food safety and food security, badly needed to feed world’s growing population but also for food waste reduction in order to save substantial amounts of greenhouse gas emissions.

## 1. Introduction

Food safety and food security are of primary importance to fight the hunger of a reported 828 million people in 2021, so to deliver the required amounts of nutritive and highly valuable food products for everybody [1]. It has been estimated by the world health organization (WHO) and others that at least about 10% of the world’s population suffer each year from severe foodborne illnesses [2,3,4]. In addition to the affected diseased, more than 40% of food is lost along the food production chain from farm to fork, causing about 10% of total greenhouse gas emissions each year. The economic burden for food waste lost was given by an almost inconceivable 2.6 trillion USD including environmental and social costs [5,6]. One effective measure to reducing food waste and food infections would be the prolongation of the shelf life of perishable food products. How can this be achieved? In wealthy countries, the infrastructure for food transportation and food storage in companies and retail facilities, mainly done by refrigeration, assures on-time delivery of food to the customer. However, cooling and freezing of food is energy-consuming and generates unwanted emissions. In less-developed countries such food logistics are simply not affordable. Hence, there are good reasons for continuous extension of more simple and cost-effective alternatives to preserve food products.

Chemical preservation is one option, which has its limitations. Food preserved with sugar, salt, or organic acids changes texture and taste or, as in the case of sorbic acid, are not acceptable by the customer, because all-natural products that contain as few food additives as possible are en vogue [7,8]. Physical preservation requires equipment and energy for heating or cooling and has its limitations as sensory characteristics can sometimes undesirably alter the food product (i.e., high-pressure processing) [7]. Biological preservation can be achieved through food fermentation by microorganisms such as yeasts and lactic acid bacteria (LAB) [9]. This has led in human history to new, often very tasty food products, accompanied by prolonged preservation resulting in shelf-life extensions from hours or days to weeks or even months. Already about 2000 years ago, the availability of wine as a safe source for drinking supply together with the increased availability of fermented foods was a factor not to be underestimated for the Roman armies to build up the Roman empire [9,10]. When in the early 20th century Ilja Iljitsch Metchnikow described the activities of non-pathogenic, health-promoting LAB as the basis of fermented milk products such as kefir and yoghurt, it became obvious that microorganisms are the indispensable actors, the so-called starter cultures [11,12]. While fermentation generally influences the organoleptic properties of food products by secreting organic acids, ethanol, carbon dioxide, and numerous aromatic compounds, they produce concomitantly small peptides, the bacteriocins, and other bactericidal compounds such as diacetyl. The observation that species from the order *Lactobacillales* can be included in food products as protective cultures to fight pathogenic and food spoiling bacteria without interfering with food taste and texture paved the way for the field of bio-protection [13]. Hence, the use of protective cultures in perishable foods such as smoked salmon, minced meat, salad, and sprouts has resulted in a significant increase in shelf life and food safety.

In this review, we address the application of LAB as food protective cultures. We describe the characteristic features of protective microorganisms regarding the underlying mechanisms. We outline the strategies for strain identification and validation relating to application in diverse food products and provide a survey on the market. For the first time, we extensively attribute the legal restrictions for the use of protective cultures to provide a guideline to the interested user.

## 2. Antibacterial Properties

The concept of bio-protection of food by the implementation of protective cultures is widely accepted, is substantiated in numerous studies, and is implemented into a wide range of products on the market [13,14,15]. As a first precondition, the activity of a protection culture in a certain food environment must leave the organoleptic properties untouched. However, there is more to consider: (i) production of antimicrobial substances should be effective; (ii) survival during the manufacturing process must be guaranteed; (iii) bacteria must be able to function at cooling temperatures; and (iv) they should be proofed as safe regarding the absence of pathogenic genes and genes encoding resistances to antibiotics. How do they exert these properties?

The addition of live bacteria to food products as protective cultures was based on the observation that bacteria have developed a number of strategies to outplace competing species. An initial observation was that a bacterial species can overgrow in co-culture experiments other species simply by the higher cell amount of the inoculum, depriving nutrients and minerals. The so-called Jameson-Effect proved to be valid in many cases [16,17,18,19,20]. Nevertheless, when alternative nutrients are available, it does not always work [21].

Microorganisms have evolved mechanisms to survive in their ecosystems. One example is the capacity of soil-dwelling *Streptomycetes* that secret bio-active secondary metabolites as a survival strategy against other microorganisms, a process that is triggered by carbon regulation [22,23,24]. Among these metabolites are more than two-thirds of medically used antibiotics [25]. Members of the order *Lactobacillales* exert as well a multitude of measures against other microorganisms (Figure 1). Through homo- and heterofermentative fermentation of carbohydrates, they produce a variety of organic acids which they secrete as the metabolic end products [26]. While lactate and acetate are quite common, there are LAB that make propionic acid, malic acid, succinic acid, butyric acid, or formic acid [14,27]. 

Organic acids are entering the cell in their uncharged form where they dissociate and lower the cell internal pH [28,29]. Most bacteria keep and regulate internal pH between six and seven as long as the external pH is in the range of five to eight [29]. Further acidification disturbs the cell internal ionic balance through the disruption of cell membranes, enzyme denaturation and damage of nucleic acids. Notably, the proton gradient across the membrane collapses which inhibits ATP Synthase, proton symporters, and the proton-driven flagellar motor (Figure 1) [27,30].

Hydrogen peroxide, carbon dioxide, and diacetyl are widely produced antimicrobial agents. LAB synthesize hydrogen peroxide in the presence of oxygen and light to amounts up to 1 mM [31,32]. Among 193 examined LAB species, 37 were able to secret hydrogen peroxide [31,33]. Bacteria that do not possess enzymes for breakdown such as peroxidases or catalases are posed under stress through this oxidative compound that target thiol groups within enzymes and exert oxidative cell reactions such as uncoupling the electron transfer chain [31,34]. *Escherichia coli* and *Salmonella enteritidis* strains that possess catalase are inhibited as well, since in this case the overall enzyme activity is often not sufficient for detoxification [31,33].

Decent amounts of the butter flavor diacetyl, formed by LAB, are present in many food products [35]. Diacetyl has been shown to inhibit food pathogenic Gram-negative bacteria, molds, and yeasts in the range of 50 to 1000 ppm [35] but has been shown to be less effective against Gram-positive bacteria [36]. It has been identified as a mutagenic agent by the Ames-Test [37,38] and attributed to the inhibition of arginine utilization [35]. It should be noted that diacetyl could be toxic for humans through inhalation of this volatile compound (Figure 1) [39].

Lactic acid bacteria that produce carbon dioxide have been shown to inhibit the growth of Gram-positive bacteria and Gram-negative psychrotrophs, including *Enterobacteriaceae* and *Listeria monocytogenes* [40]. Carbon dioxide can alter bacterial communities, reduce metabolite production, and as an example improve in this way the quality of fermented kimchi [40,41]. The mechanisms comprise the creation of an anaerobic atmosphere, the decrease of pH accompanied with enzyme denaturation, and the loss of the proton gradient and cell membrane [42]. Secreted aldehydes also confer antimicrobial function. The best studied one is reuterin (3-hydroxy propionaldehyde) from *Lactobacillus reuteri* strains. It causes oxidative stress and interacts with thiol groups [43]. All groups of microorganisms including spores and protozoa are targeted by reuterin [44].

Of central importance is the production of numerous bio-protective peptides, the bacteriocins (Figure 1) [45,46]. The class I contains peptides with less than 10 kDa that are posttranslational modified and thus have unusual amino acids [47]. In class II, non-modified small bacteriocins are classified. They are independent from modifying cell enzymes. Peptides bigger than 10 kDa, all unmodified, are in class III. By bacteriocin-mediated pore formation, the cell wall is targeted, and damage occurs in the peptidoglycan layer [48]. Bacteriocins bind to cell envelope structures like Lipid II or to the PEP:dependent mannose phosphotransferase transport system [49]. Some bacteriocins act on the central cell components DNA, RNA, and protein metabolism (Figure 1) [45,46]. Diverse bacteriocins can exhibit a very narrow spectrum against a few species or a broad spectrum against bacteria. Some of the most notable ones are nisin, produced by *Lactococcus lactis*, pediocin, synthesized by *Pediococcus* strains, or zoocin A, a product from *Streptococcus equi subsp. Zooepidemicus* [50]. The screening for genes and operons encoding bacteriocins has been facilitated by the BAGEL4 database server [51]. In 2018, there have been 820 gene loci detected in 238 genomes, best studied in the genera *Lactobacillus* and *Streptococcus* [47].

Although much knowledge has been acquired through the past decades on the above-mentioned antibacterial compounds and bacteriocins, there are still research gaps to fill in that require further research efforts (Figure 1). The molecular mechanisms of the numerous antibacterial actions are still to some extent a mystery and thus await discovery. In addition, the invention of high throughput technologies to directly identify novel bacteriocins is still a drawback. However, the recent launching of a micro-bioreactor could help solving this, since the device allows parallel fermentation of 48 cultures in 0.8 mL to 2.4 mL [52].

Regarding application, there are many studies demonstrating that protective cultures can be applied in various food to enhance food safety [53,54,55,56,57,58,59,60,61,62,63,64]. A few selected studies in various food groups are given here. Meat products made of cooked ham and minced meat are most sensible towards spoilage and contamination with pathogenic bacteria. Cooked cubed ham was inoculated with a mixture of *L. monocytogenes* strains and was challenged with two commercial protective cultures, Lyocarni BOX-74 and Lyocarni BOX-57. Storage for 40 days at a household-representative refrigerator temperature of 8 °C revealed that *Listeria* have been eliminated without organoleptic changes from the ham, making this setting a proof-of-concept for cooked ham products [53]. Stephane Chaillou and co-workers applied *L. sakei* strains to ground beef stored at 4 °C and 8 °C, respectively. A protective effect could be demonstrated against *Salmonella enterica Typhimurium* and *Escherichia coli* O157:H7 [54]. Among marine products, smoked salmon is of particular safety concern due to contamination with *L. monocytogenes* and also due to a number of strong-smelling spoilage bacteria like *Photobacterium phosphoreum*, *Brochothrix thermosphacta*, and *Serratia proteamaculans*. It was shown in several publications that appropriate LAB could stabilize such products [55,56]. Fresh cheese has been studied by the isolation and investigation of eight LAB strains. A mixture of two strains demonstrated great potential as a protective culture for the cheese making process against listerial contamination [57]. Fruits and vegetables can be contaminated predominantly with *E. coli* EHEC ssp., *Salmonella enterica* ssp., or *L. monocytogenes*. Since such food is consumed raw, numerous outbreaks have happened in past decades [58,59]. The functionality of the prevention of pathogens has been shown for cut fruits, pre-cut cantaloupes, and papayas as well as in iceberg salad, cucumber, or romaine lettuce [58,60]. A quite novel approach for application is the improvement of food safety in catering by protective cultures, especially concerning vulnerable groups. By spraying protective LAB on certain dishes before serving, the risk of food infections was significantly reduced [61]. The above-mentioned selection of thorough investigations shows the potential for food produced with protective cultures to improve food safety. We will further address this on the perspectives of the market.

## 3. Isolation and Validation of protective LAB and Bacteriocins

LAB are ubiquitous present in the environment and thus can be isolated from a wide variety of food-associated habitats such as milk from buffalo, cow, and goat and their respective farm environments. A good source are wild fermented food products like cheese, sausages, or wine. These are made by taking advantage of the autochthonic flora. Other suitable sources are the gastrointestinal tracts of fishes or insects. Even from honeydew and pig feces, the isolation of potential new protective cultures or possible bacteriocin producers has been demonstrated [65,66,67,68,69,70]. Samples are plated onto De Man, Rogosa, Sharpe Agar (MRS), or Brain Heart Infusion Agar (BHI), sometimes supplemented with growth-stimulating additives such as cysteine, fructose, or bromphenol blue (BPB) [67,70]. New species can be readily distinguished by colony morphology in shape, size, and color. In our hands the best choice is MRS-BPB agar as a chromogenic differential medium that delivers all kind of blueish colonies (our unpublished results; Figure 2A [71,72,73].

If new strains are available as pure cultures, species identification is conventionally performed by 16S-rRNA sequencing [67,69,70,74]. However, due to the conserved behavior of the 16SrRNA gene, unambiguous identification at the species level will not always be possible. Instead, PCR amplification with primer pairs corresponding to less-conserved genes will be the solution. It should be noted that only species can be identified for which the genome has been sequenced [75,76]. Alternatively, species identification can be achieved by comparison of the in silico-generated proteolytic protein fragment pattern using matrix-assisted laser desorption ionization time-of-flight mass spectrometry (MALDI-TOF MS). Reliable identification with this rapid detection method is particularly dependent whether the respective species has been sequenced and deposited in the database [66,70].

In a next step, the isolate is characterized in terms of antimicrobial properties and biological safety. This is addressed by (i) in vivo detection of antibacterial activity against indicator strains or (ii) by genome sequencing followed by bioinformatic searches for the presence of bacteriocin-encoding genes. For detection in vivo, the point inoculation assay (Figure 2B), the spot-on-lawn assay, or the agar-well diffusion assay is available [69,74,77,78,79,80]. These assays can also be used to clarify a sporocidal or sporostatic effect [65,81]. Since a response against an indicator organism does not necessarily have to be based solely on the action of bacteriocins, possible effects of acidification or hydrogen peroxide should be clarified first. The production of the latter can be analyzed with a 3,3’,5,5’-tetramethyl-benzidine (TMB) and catalase-containing MRS agar, in which a color change of the colony from white to dark blue occurs if hydrogen peroxide is produced (Figure 2C). In this case, the produced amounts should be quantitatively determined [82,83]. Secretion of acid can be measured with pH indicator stripes placed on the agar plate as shown in Figure 2D. The tolerance of the isolate to low pH environments (acidification) can be determined in a survival challenge test by incubation at a low pH-range between two to five [66].

In order to meet the required criteria for food safety, it is necessary to show the absence of antibiotic resistant genes and genes encoding virulence factors [66,84,85]. It is recommended to substantiate the analyses in vivo by antibiotic susceptibility testing [84]. A convenient and reliable strategy is whole genome sequencing, which is cost- and time-efficient since the invention of nanopore sequencing technology, followed by bioinformatics [51,86,87,88,89,90,91]. A versatile tool is the BAGEL4 webserver that has been established within the group of Jan Kok and Oscar Kuipers at the Groningen Biomolecular Sciences and Biotechnology Institute (GBB), Netherlands [51]. The database comprises 820 genes encoding bacteriocins [51].

While the examination of the safety criteria is ongoing, experiments could be conducted in parallel to implement the new isolated protection strain into the chosen food product. The microbiological profile should be followed to determine the shelf-life. Absence or prevention against potential food pathogens, texture analysis, and proof of unchanged organoleptic properties will complete characterization. Subsequently, the upscaling process must follow in order to commercially sell the food product in sufficient quantities.

## 4. Industrial Production of Protective Cultures, Bacteriocins, and Their Actual Imitations

The industrial production of protective cultures in a bioreactor follows those for LAB used as food starter cultures. The protective cultures are cultivated from a stock culture using small-volume batches of about 2 mL to 100 mL or a fed-batch process to grow the bacteria until a desired biomass concentration is reached. The biomass serves to inoculate 200 L to 50,000 L bioreactors in a batch or fed-batch process [92]. At this stage, bioreactors no longer work with defined media, but instead with cheap raw materials such as sidestream products from cereals or sugar beets to minimize costs and maximize yield (Table 1). Decisive for a high biomass during fermentation is keeping the culture in the exponential growth phase and within the appropriate environmental parameters. They include oxygen content, redox potential, stirring speed, pH, and temperature. Although LAB are acid tolerant, self-acidification primarily by lactic acid has a more negative influence on biomass formation than limitation of nutrients. Controlling pH at a value close to neutral is critical for ensuring highest growth rates and a maximum of biomass [93]. 

A compromise between protective culture as maximum biomass and pure bacteriocin production are fermentates. Fermentates are the bacterial cells propagated in the reactor plus medium into which bacteriocins are secreted. Under optimally adapted physical and chemical conditions in vitro, LAB are able to secrete bacteriocins into significantly higher amounts than in vivo in a food matrix [98]. However, many factors influence the secretion of large amounts of bacteriocin into the medium, for example pH [99,100], temperature [100], nutrient density, and salt content. The higher the salinity the lower the yield of the respective bacteriocin [101]. On the other hand, salt stress may also be necessary to secrete bacteriocins [102]. This poses complex challenges between biomass and bacteriocin production. Especially, the focus on bacteriocin production should result in a yield of a minimum of 50% and a purity of 90% [103]. In most cases, the yield is significantly lower and is then associated with large efforts to improve culture conditions and purification process [104,105].

An alternative approach for bacteriocin production is the molecular biology approach in which bacteriocin-encoding genes or operons are cloned into strains that are widely used in biotechnology. Since the gene sequences of numerous bacteriocins are readily available [68,106], the successful construction of recombinant bacteriocin-producing *Escherichia coli* and *Corynebacterium glutamicum* has been demonstrated [107,108,109,110,111].

## 5. The Legal Use of Protection Cultures and Bacteriocins in the European Union

The use of microbial protective cultures is not generally regulated as an ingredient, additive, or processing aid in the regulatory framework of the European Union (EU) with its 27 member states and the European Economic Area (EEA) and does not necessarily require approval and safety assessment. The duty of care and the sole responsibility for the specific use lies for the time being solely by the user. Further regulation is carried out outside the legal requirements of the Basic Food Regulation (EC) No. 178/2002 [112] on the basis of its technological purpose. Only Denmark and France are exceptions and register microbiological cultures on a national level as additives [113,114], regardless of their intended use, and require in part proof of efficacy and safety in addition to notification and approval. However, this does not reduce the free market access of foods treated with protective cultures from other EU countries.

In addition, food- or feed-associated microorganisms must always undergo a safety assessment and be approved by the European Food Safety Authority (EFSA) [115] in the EU if they are considered genetically modified (GMO) by Directive 2009/41/EC [116], Regulation (EC) No 1829/2003 [117], and Directive 2001/18/EC [118]. They are classified as a novel food by Regulation (EU) No 2015/2283 [119] or are considered a feed additive according to Regulation (EC) No 1831/2003 [120]. In 2006, the EU Standing Committee on the Food Chain and Animal Health proposed that microbial cultures with a technological purpose, such as food preservation, should be considered as additives [121], which would require safety assessment and approval of preservative cultures by the EFSA. This turns out to be problematic if they originally belong to or have developed from starter cultures that are not specifically regulated in the EU. Starter cultures also have the additional benefit of reducing the potential risk of pathogenic microorganisms. Therefore, a differentiation must be made according to the basic principle of the effectiveness of the protective culture.

The effect is based on displacement cause by microorganisms that have a higher colonization rate. They can outcompete nutrients through more efficient transport systems of carbon sources or by an accelerated metabolism. They also may exert superior adaptation to aerobic or anaerobic, or other growth conditions. In such cases, the activity is attributed to a non-specific protective culture [19,122,123]. This also applies to the antagonistic formation of acids such as lactic, acetic, benzoic, malic, succinic, and formic acid or ethanol, diacetyl, hydrogen peroxide, and carbon dioxide [29,35,40,124].

Specific protective cultures are distinguished from non-specific ones by their technological purpose and their action by means of bacteriocins, proteins, or protein-like compounds, such as nisin, with antagonistic activity against defined foodborne pathogens [75]. This distinction results from the European Food Additives Regulation (EC) No 1333/2008 [125] and the prohibition of the use of unauthorized substances in food according to Article 5 of Regulation (EC) No 1333/2008 [125].

The prohibition does not refer to the microorganisms themselves but to the substances produced by them. This is further differentiated in the Enzyme Regulation (EC) No. 1332/2008 [126]; the exemption from approval and regulation does not apply to the use of microorganism cultures for the targeted production of substances with a technological effect. LAB cultures, for example, can be used without prior approval as protective cultures that produce not only lactic acid but also the bacteriocin nisin (E234), which is approved as a food additive. However, the maximum permissible quantities for nisin must not be exceeded and nisin must be declared accordingly (Table 2). Apart from nisin, there are no other bacteriocins permitted as food additives in the Additives Ordinance as amended on 31 October 2022. A distinction between non-specific and specific protective cultures is not necessary for all microbial cultures; if they were not contained in foodstuffs in the European Union before 14 May 1997 [127] and were marketed in significant quantities for consumption, they all fall under the novel food Regulation (EU) No 2015/2283 [119] and require a marketing authorization.

If an application for market authorization and with it a safety assessment on the part of EFSA is required, there are two approaches depending on the organism used. In principle, the application procedure for food additives is carried out in accordance with Regulation (EC) No 1331/2008 [128]. The technical and administrative requirements for an application to be submitted are set out in Regulation (EC) No 234/2011 [129]. In addition, EFSA provides practical guidelines to support the submission of an application for marketing authorization [130].

The two approaches for safety assessment differ in terms of their depth of testing. If the EFSA already has a qualified presumption of safety (QPS) status for the microorganism used, an accelerated generic safety assessment will be performed. This procedure was introduced by the EFSA in 2007 to simplify the marketing authorization process and save resources. For a microorganism to have QPS status, the taxonomic unit of the organism used must be at species level for bacteria, fungi, and microalgae/protists and at the family level for viruses, and it must be on the QPS list [131] maintained by the EFSA. This QPS list is reviewed by the EFSA Panel of Biological Hazard (BIOHAZ) and updated every six months. 

If a microorganism is used that does not have QPS status, there may be two reasons: First, the organism has never been evaluated because no dossier was ever submitted for authorization and therefore the QPS evaluation process was not initiated by EFSA. Second, the micro-organism in question was not granted QPS status following a QPS evaluation by the BIOHAZ Panel [132]. The QPS workflow by the BIOHAZ panel is shown in Figure 3. This can happen if the taxonomic status of the microorganism is ambiguous, if there is not sufficient study evidence for the microorganism, and/or if the microorganism used has potentially harmful characteristics such as pathogenicity, antibiotic resistance, or virulence factors or produces biologically active toxic secondary metabolites. However, this does not constitute a disqualification for marketing authorization, as a full safety review is foreseen for these organisms. The 2017 work by Laulund and colleagues describes possible tools that can be used to review a list of methods and procedures for assessing the safety of food cultures at the strain level and as quality assurance measures during production and in the final product [133]. These can be applied to protective cultures as well. An overview of all microorganisms, notified for marketing authorization since 2007, can be found on the list of microbiological agents as notified to the EFSA [134].

## 6. The Legal Use of Protection Cultures and Bacteriocins in the United States of America

In contrast to the partially undefined European legal framework for microbial cultures, the Federal Food, Drug, and Cosmetic Act [136] has provided a corresponding legal framework in the United States of America (US) since 1958, which was revised in 1997. Both legal frameworks are fundamentally different, even if at first glance there are two similar programs with the GRAS program [137] (generally recognized as safe) of the U.S. Food and Drug Administration (FDA) and the QPS list at the EFSA.

In the US, according to the Federal Food, Drug, and Cosmetic Act [136], a food additive is basically any substance that directly or indirectly becomes a component of a food or could influence the food in one way or another. Thus, unlike in the European Union, the term food additive is very broad and encompasses virtually as what comes into contact with food of a pre-approval by the FDA. The only exception is that it is a substance that has been recognized by the FDA generally recognized as safe (GRAS). Accordingly, microbial protective cultures, microbially derived ingredients such as bacteriocins can be introduced to the market either as a food additive or by obtaining GRAS status. While a food additive requires an evaluation and approval by application according to 21 CFR Part 171 [138], this is not necessary to obtain GRAS status. Here, the regulations according to 21 CFR 170.30 [139] apply, which contain some special features. The GRAS status is provided for one microorganism at a strain level and one specific application. This means that an additional GRAS status must be provided for each additional use case.

In addition, the GRAS status is not determined by the FDA, as is the case with food additives, but by qualified experts. The first distributor or manufacturer is free to decide how the expert panel is convened and staffed. An evaluation can also be carried out by the FDA if desired. The Guidance for Industry [140] assists in the implementation of the requirements under 21 CFR 170.30 [139]. If there is a GRAS conclusion by the expert panel, the marketer may make a notice of congestion to the FDA under 21 CFR 170 Subpart E [141]. However, the marketer or manufacturer is not required to do so under either the Federal Food, Drug, and Cosmetic Act [136] or 21 CFR 170.30 [139,140]. Upon submission of the GRAS conclusion, the FDA either agrees with the conclusion, and the GRAS status exists further, or denies it. However, even then it is not prohibited to use the particular culture in food under U.S. law unless the FDA issues a ban. In the event of a claim, the liability issues are different than with GRAS status granted by the FDA. Table 3 shows the main difference between the GRAS system and its European counterpart QPS.

## 7. The Legal Use of Protection Cultures and Bacteriocins in Latin America

In Mexico, the use of microbial protective cultures in food products is regulated by the Federal Commission for Protection against Sanitary Risks (COFEPRIS) under the Ministry of Health. The use of these cultures is permitted only when they are considered safe for human consumption and do not alter the nutritional value of the food product. The manufacturer of the protective culture must provide evidence that the culture is safe and effective for use in the specific food product. This requires GRAS status on the part of the FDA or its homologation. Further, a sample of the microbiological culture used must be deposited in the strain collection for microorganisms of the National Institute of Forestry, Agriculture and Livestock Research (INIFAP) [142].

The use of microbial protective cultures in Brazil is regulated by the National Health Surveillance Agency (ANVISA). ANVISA requires that manufacturers of protective cultures provide evidence that the culture is safe and effective for use in food products. Since the revision of Resolution RDC 27/2010: Categories of food and packaging that require pre-market approval by Anvisa by Resolution—RDC No. 240, of 26 July 2018 the registration and approval of microbial cultures as technical adjuvants is no longer required [143].

In Peru, the General Directorate of Environmental Health (DIGESA) of the Ministry of Health is responsible for regulating the food industry. Currently, there is no regulatory framework for the use of microbial protection cultures in food.

## 8. The Legal Use of Protection Cultures and Bacteriocins in Asia

The regulatory framework for microbial-protective cultures in China is primarily governed by the Food Safety Law, which sets the basic principles for food safety, and the Regulations for the Administration of Food Additives, which provides specific requirements for the use of food additives, including microbial protective cultures. The regulatory authorities in China are the National Health Commission and the Department of Food Safety Standards, Monitoring and Evaluation, which since 2010 has had a separate regulation for food cultures, including protective cultures. From this date, all new strains placed on the market must be on the official list. The list was last updated on 18 August 2022 [144].

Some other countries have also established positive lists for food and protective crops. Examples include Thailand and Malaysia. In Thailand, industrially used cultures are considered food additives. The regulatory framework requires the submission of strain level documentation for registration with the Food and Drug Administration [145]. Traditional fermented foods (spontaneous fermentation) without the use of food cultures are not subject to registration. Since 2014, Malaysia has provided a “New Regulations 26B” with its Malaysian Food Act, which provides the regulatory framework for “Microbial Cultures for Food Fermentation”. Protective cultures also fall under this regulation [146].

## 9. Market Overview

The economic importance of protective cultures for food processing should be appreciated. The global market volume in 2022 was already 1.3 billion USD [147] in the food and beverage sector. Consumers now expect not only safe food but also healthy food that is as natural as possible [148]. In addition, social goals such as sustainable ecological management and the reduction of food waste are increasingly prompting food companies to take action. Companies can meet these challenges through the innovative use of microbial protection cultures. Several studies have shown that the use of protective cultures on food can extend the shelf life of products such as fish or meat, thus avoiding food waste [56,75,149,150,151,152]. The potential elimination of refrigeration contributes to carbon reduction and sustainability goals and enables safe food in regions without refrigeration infrastructure. Prolonged storage of grapes (12 days) and meat (2 days) at room temperature, rather than in refrigeration, after treatment with a protective culture or its ferment, demonstrated the potential benefits in this case [153,154]. In addition, the elimination of artificial preservatives brings healthier and more natural products to the market. Consequently, there is a great development and growth potential for protective cultures in the food sector, which market research companies estimate to have an annual growth rate between 4.3% and 23.3% by the end of the decade [147,155,156]. Market development is not only driven by further growth in Europe and North America but also by the Asian market, which is home to 60% of the world’s population and is developing consumer needs comparable to those in Europe and the US [147,155].

However, the market for protective cultures is quite fragmented and is not dominated by large international companies. In addition to finished products, start-ups and medium-sized companies in particular develop customer- and product-specific cultures [147,155]. Protective cultures are primarily offered as freeze-dried or frozen preparations and as ferments. The only bacteriocin used in pure form is nisin. Table 4 shows a compilation of ready-to-use commercial protective cultures available in the European Union categorized by food application. As shown in Figure 4, almost 60% of the listed commercial protective cultures address a single category, that of milk and dairy products, and currently represent the largest area of application for protective cultures.

## 10. Conclusions

The implementation of protective cultures of lactic acid bacteria in diverse perishable food products contributes substantially to food protection, food safety and food security. The here thoroughly discussed regulatory frameworks show that the use of foods harboring protective cultures is superior to the application of specifically defined agents, such as a pure bacteriocins. This accounts even more regarding the production costs. The current state of knowledge through studies providing proof of principle for all categories of risky foods shows that a wealth of protective cultures is available. Furthermore, nature still has an untapped unlimited reservoir for more protective bacterial species that remain to uncover. Bio-protection could thus contribute to nourish the hundreds of millions of starving people and to reduce significantly greenhouse gas emissions by food waste reduction. The biggest challenge to making this true is probably to speed up the transfer of scientific knowledge towards food production companies.

## Figures and Tables

**Figure 1 foods-12-01541-f001:**
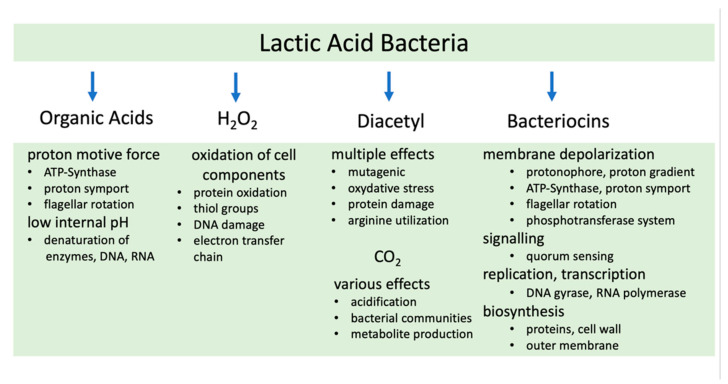
Mode of antibacterial actions by LAB protective cultures. Shown are the major molecules that exert antibacterial activity and the identified targets.

**Figure 2 foods-12-01541-f002:**
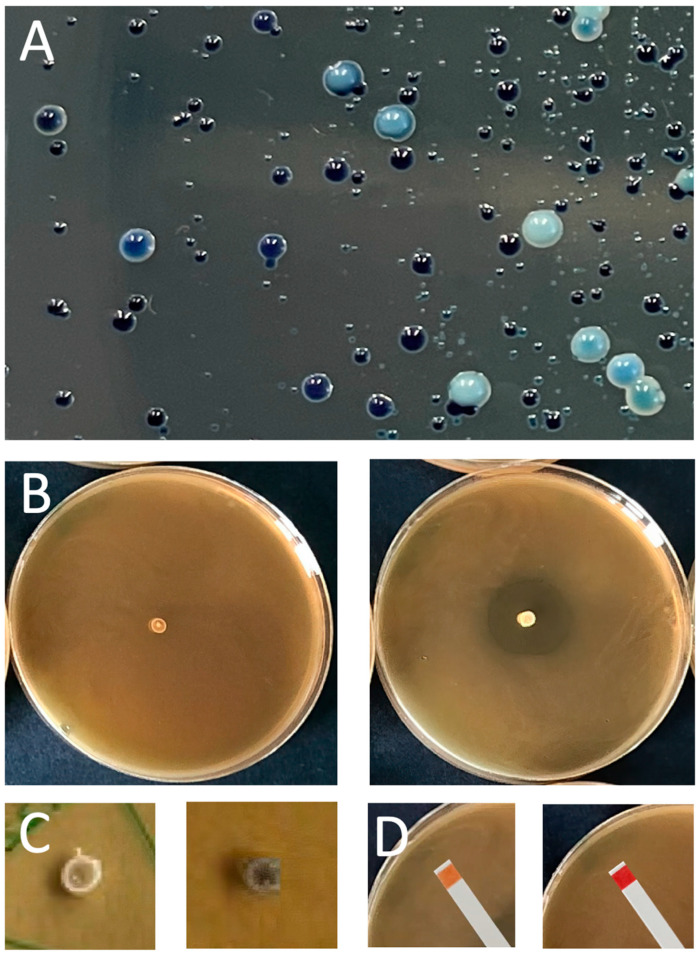
Lactic acid bacteria: (**A**) Diverse blueish colonies of LAB species on mMRS-BPB differential agar. (**B**) In point inoculation assay of LAB (center colony) against a pathogenic indicator strain (bacterial lawn). Left without zone of inhibition (no bacteria-killing activity), right with zone of inhibition (bacteria-killing activity). (**C**) Detection of hydrogen peroxide with TMB; left, light colony showing no secretion; right, blue colony indicating secretion. (**D**) Detection of acid production by LAB (MRS agar plates); pH stripes: orange, acidification to pH 4.1; red, no acidification pH 6.1.

**Figure 3 foods-12-01541-f003:**
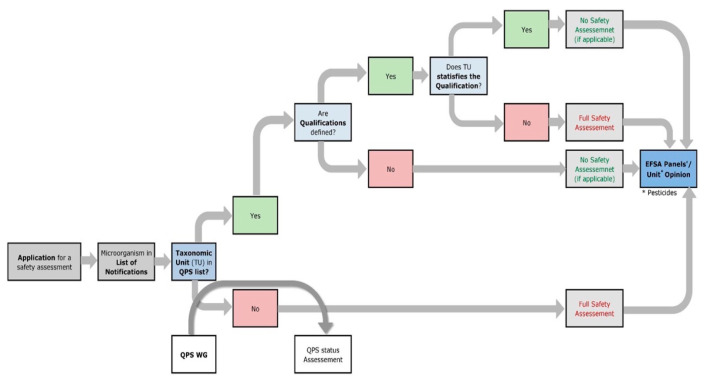
Workflow diagram describing how EFSA Units incorporate the qualified presumption of safety (QPS) status (from Herman et. al. under CC BY 4.0 conditions) [135].

**Figure 4 foods-12-01541-f004:**
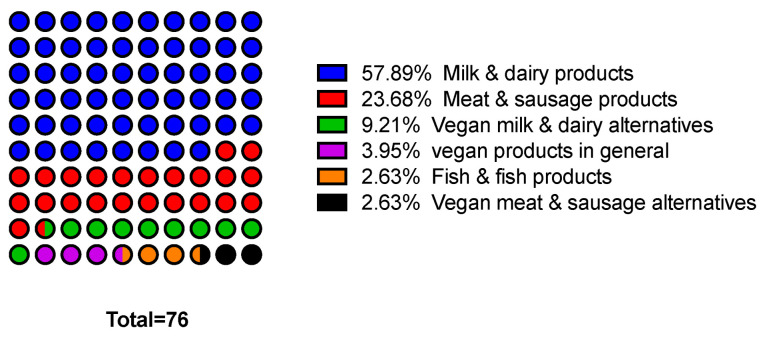
Commercially available protective cultures by food application.

**Table 1 foods-12-01541-t001:** Ingredients used in industrial growth media [94,95,96,97].

Ingredient	Source
carbohydrate	glucose or lactose (skim)milk or whey cellobiose–xylose mixture) sugar molasses
amino acids or peptones	mainly added though peptones, yeast extracts or proteins (milk)
fatty acids	tween, acetate
nucleosides/nucleotides	yeast extracts, meat
minerals	mainly added as salts
vitamins	yeast extracts, meat extracts, and peptones
citrate	purified dairy-based ingredient

**Table 2 foods-12-01541-t002:** Allowed addition of nisin in food, natural occurrence in the products not included, source Regulation (EC) 1333/2008.

Category	Category No.	Maximum Quantity	Restriction/Place of Use
other cream products	01.6.3	10 mg/L bzw. mg/kg	clotted cream products
unripened cheese, (except products of category 16)	01.7.2	10 mg/L or mg/kg	mascarpone
ripened cheese	01.7.2	12.5 mg/L or mg/kg	
processed cheese	01.7.5	12.5 mg/L or mg/kg	
cheese products (except category 16)	01.7.6	12.5 mg/L or mg/kg	ripened and processed products only
eggs and egg products processed	10.2	6.25 mg/L or mg/kg	pasteurized liquid only (egg white, yolk or whole egg)
dessert foods, excluding products of categories 1, 3 and 4	16	3 mg/L or mg/kg	semolina and tapioca puddings and similar products only

**Table 3 foods-12-01541-t003:** Differences between the GRAS guidelines (FDA, USA) and the QPS system.

GRAS	QPS
food additives and microorganisms	microorganisms only
FDA or external experts determine GRAS status	EFSA determines QPS status
open to all types of food additives	limited to microorganisms associated with regulated food and feed products only
applicants request a GRAS status	EFSA requests evaluation of new taxonomic units within the scope of an internal mandate
describes a specific substance or microorganism at the strain level	describes microorganisms at the taxonomic unit level
case-by-case safety assessment at the strain level	general safety assessment at the taxonomic unit level.
open tool to all applicants	internal tool by EFSA

**Table 4 foods-12-01541-t004:** Commercially available protective microbial cultures for various applications.

Food Application	Product Name	Microorganism	Manufacturer
Fish & fish products	Lyoflora FP-18	*Carnobacterium* spp.	SACCO
	Lyoflora FP-50	*Carnobacterium* spp.	SACCO
Meat & sausage products	AVO raw sausage protective culture	*Lactobacillus plantarum*	AVO-Werke August Beisse GmbH
	Lyocarni BMX-37	*Carnobacterium* spp., *Lactobacillus sakei*	SACCO
	Lyocarni BOM-13	*Carnobacterium* spp., *Lactobacillus sakei*	SACCO
	Lyocarni BOX-74	*Carnobacterium* spp., *Lactobacillus sakei*	SACCO
	Lyocarni BXH-69	*Carnobacterium* spp., *Lactobacillus sakei*	SACCO
	Lyoflora FP-18	*Carnobacterium* spp.	SACCO
	Lyoflora FP-50	*Carnobacterium* spp	SACCO
	M-Culture RS 103	*Lactobacillus curvatus* subsp. *curvatus*, *Lactobacillus plantarum*, *Lactobacillus sakei*, *Staphylococcus carnosus*	M-Foodgroup GmbH
	M-Culture RS 107	*Lactobacillus curvatus* subsp. *curvatus*, *Lactobacillus plantarum*, *Lactobacillus sakei*, *Staphylococcus carnosus*	M-Foodgroup GmbH
	M-Culture RS 38	*Lactobacillus curvatus* subsp. *curvatus*, *Lactobacillus plantarum*, *Pediococcus pentosaceus, Staphylococcus carnosus*, *Staphylococcus xylosus*, *Debaryomyces hansenii*	M-Foodgroup GmbH
	M-Culture RS 49 FM	*Debaryomyces hansenii*, *Lactobacillus sakei*, *Lactobacillus plantarum*	M-Foodgroup GmbH
	M-Culture Safe GDS 3349	*Leuconostoc Carnosum*	M-Foodgroup GmbH
	ProtectSTART	*Staphylococcus xylosus*, *Staphylococcus carnosus, Leuconostoc citreum*	Moguntia food group
	SaferSTART	*Leuconostoc citeum*	Moguntia food group
	Protective cultures for raw sausage	*Lactobacillus sakai*, *Pediococcus pentosaceus*, *Staphylococcus xylosus, Staphylococcus carnosus*, *Debaryomyces hansenii*	Würzteufel GmbH
	Protective culture Ham-Protect	*Lactobacillus plantarum, Staphylococcus carnosus*, *Staphylococcus xylosus*	Würzteufel GmbH
Meat & sausage products	Protective culture PROTECT ONE	*Staphylococcus xylosus*, *Staphylococcus carnosus*, *Lactobacillus plantarum bac+*	Würzteufel GmbH
	StartStar SAFE Protective Cultures	unknown culture with Bacteriocin producer	Holkof GmbH
Milk & dairy products	AC line DY 4P13	unknown mixture of LAB	SACCO
	AC line LC 4P1	unknown mixture of LAB	SACCO
	AC line LCP 4P2	unknown mixture of LAB	SACCO
	AC line MO K4P04	unknown mixture of LAB without *Lactococcus lactis*	SACCO
	AC line MO L4P03	unknown mixture of LAB without *Lactococcus lactis*	SACCO
	AC line MO N4P01	unknown mixture of LAB with *Lactococcus lactis* as nisin producer	SACCO
	AC line MO N4P02	unknown mixture of LAB with *Lactococcus lactis* as nisin producer	SACCO
	AL line CNBAL	unkown mixture of LAB with *Carnobacterium divergens* V41 and *Carnobacterium piscicola* SF668	SACCO
	AL line LPAL	unknown mixture of LAB	SACCO
	AOSM line BGP1	unknown mixture of LAB	SACCO
	AOSM line LR B	unknown mixture of LAB with *Lactobacillus rhamnosus*	SACCO
	AYM line CLP C	unknown mixture of LAB	SACCO
	AYM line LPR A	unknown mixture of LAB	SACCO
	AYM line LR B	unknown mixture of LAB with *Lactobacillus rhamnosus*	SACCO
	AYM line LR4 PD	unknown mixture of LAB	SACCO
	BIOPROX CP63	*Lactobacillus rhamnosus, Lactiplantibacillus plantarum*	bioprox
	BIOPROX L135	*Lactococcus lactis*	bioprox
	BIOPROX P 83	*Lactiplantibacillus plantarum*	bioprox
	BIOPROX P 94	*Lactobacillus plantarum* subsp. *Plantarum*	bioprox
	BIOPROX RP 80	*Lactobacillus rhamnosus, Lactiplantibacillus plantarum*	bioprox
	BIOPROX SP 86	*Limosilactobacillus fermentum*	bioprox
	BIOPROX Z100	*Lactiplantibacillus plantarum*	bioprox
	BS-10	*Lactococcus lactis subsp. lactis*	CHR Hansen
	FreshQ 11	*Lactobacillus rhamnosus*	CHR Hansen
	FreshQ 2	*Lactobacillus rhamnosus*	CHR Hansen
	FreshQ 4	*Lactobacillus rhamnosus*	CHR Hansen
	FreshQ Cheese 1	*Lactobacillus rhamnosus*	CHR Hansen
	HOLDBAC GP10	*Pediococcus acidilactici*	IFF
Milk & dairy products	HOLDBAC GP20 FRO 500DCU	*Lactococcus lactis* subsp. *lactis (nisin producer)*, *Lactobacillus paracasei, Lactobacillus plantarum*	IFF
	HOLDBAC GP21 FRO 500DCU	*Lactococcus lactis* subsp. *lactis (nisin producer)*, *Lactobacillus rhamnosus*, *Lactobacillus plantarum*	IFF
	HOLDBAC LC	*Lactobacillus rhamnosus*	IFF
	Holdbac LC Lyo 100 DCU—Schutzkultur	*Lactobacillus rhamnosus*	DANISCO (IFF)
	HOLDBAC Listeria	*Lactobacillus plantarum*	IFF
	Holdbac YM-B	*Lactobacillus rhamnosus, Propionibacterium freudenreichii* subsp. *Shermanii*	IFF
	Holdbac YM-B LYO 100 DCU—Schutzkultur	*Lactobacillus rhamnosus, Propionibacterium freudenreichii* subsp. *shermanii*	DANISCO (IFF)
	HOLDBAC YM-C	*Lactobacillus paracasei, Propionibacterium freudenreichii* subsp. *Shermanii*	IFF
	HOLDBAC YM-C PLUS	*Propionibacterium freudenreichii, Lactobacillus paracasei*	IFF
	HOLDBAC YM-SUSTAIN	*Lactobacillus plantarum, Lactobacillus paracasei*	IFF
	HOLDBAC YM-XPK	*Lactobacillus plantarum*	IFF
	HOLDBAC YM-XPM	*Lactobacillus plantarum, Lactobacillus paracasei*	IFF
	HOLDBAC YM-XTEND	*Lactobacillus plantarum, Lactobacillus rhamnosus*	IFF
	LC|100 DCU	*Lactobacillus rhamnosus*	DANISCO (IFF)
	Pure Appeal 01	*Lactobacillus paracasei, Streptococcus thermophilus*	CHR Hansen
	YM-B|100 DCU	*Lactobacillus rhamnosus, Propionibacterium freudenreichii* subsp. *Shermanii*	DANISCO (IFF)
vegan products in general	HOLDBAC YM-VEGE	*Propionibacterium freudenreichii* subsp. *shermanii, Lactobacillus rhamnosus*	IFF
	Vega FreshQ 101	*Lactobacillus rhamnosus*	CHR Hansen
	Vega SafePro 01 data	*Lactobacillus sakei*	CHR Hansen
Vegan meat & sausage alternatives	Safe M-Culture Vegan	*Leuconostoc carnosum*	M-Foodgroup GmbH
	Protective culture PROTECT ONE	*Staphylococcus xylosus, Staphylococcus carnosus, Lactobacillus plantarum bac+*	Würzteufel GmbH
Vegan milk & dairy alternatives	Lyofast BGP 1	unknown mixture of LAB	SACCO
	Lyofast CLP C	unknown mixture of LAB	SACCO
Vegan milk & dairy alternatives	Lyofast CNBAL	unknown mixture of LAB with *Carnobacterium divergens* V41 and *Carnobacterium piscicola* SF668	SACCO
	Lyofast LPAL	unknown mixture of LAB	SACCO
	Lyofast LPR A	unknown mixture of LAB	SACCO
	Lyofast BGP 1	unknown mixture of LAB with *Lactobacillus rhamnosus*	SACCO
	Lyofast CLP C	unknown mixture of LAB	SACCO
	Lyofast LR B	unknown mixture of LAB	SACCO
	Lyofast LR4 PD	unknown mixture of LAB	SACCO

## Data Availability

The data presented in this study are available on request from the corresponding author.

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
