# Peer review of "Protective Cultures in Food Products: From Science to Market"

_foods, 2023, doi:10.3390/foods12071541_

Round 1

Reviewer 1 Report

The review, “Lactic acid bacteria as protective cultures in food” focuses on the overview of lactic acid bacteria in food formulations.  The study is well presented and contributes to the field. In my opinion, below suggestions should be considered in this manuscript.

Abstract is clearly presented

Citation format must be corrected according to foods format

Line 39: There are many short sentences. Authors must re-write it throughout the manuscript.

Line 44: why not using….. authors must use academic language in writing a review article. I suggest authors to revise the entire manuscript in a more scientific and academic way

Introduction:

The need of LAB in formulation of novel foods must be included. Why do we still need LAB in food formulations? How it is related to food security and eco-friendly must be addressed

2. Protection cultures and bacteriocins: provide a graphical representation of this section as a Figure

Line 156: stains or strains ?

In some places authors used very odd language. I suggest to revise it

5. The legal use of protection cultures and bacteriocins in the European Union: I suggest to provide a global perspective as this culture is widely accepted. Every country has certain laws.

Provide the global perspectives and laws

Figure 1.: quality must be improved

Table 4: copyright permission is needed?

References must be cross-checked and format according to the journal guidelines. 

Reviewer 2 Report

The manuscript can be considered for publication after the following issue has been addressed:

-Some references cited are very old. i.e. Reference no. 16 year 1962. Please cite the latest source of the information appropriately. 

- Please improve on the English language and style of writing. 

-There are papers about appraisal of lactic acid bacteria as protective cultures have been published. Please revise the title and the contents to provide clarity, novelty, and contribution to the area.

-Please add one interesting figure about the mechanisms of the underlying mode of action or antibacterial mechanisms of the protective cultures. Also, please focus more in-depth discussion on the antibacterial mechanisms. I.e., Line 106-107 the authors wrote "The underlying mechanisms comprise the creation of an anaerobic atmosphere, decrease of pH accompanied with enzyme denaturation, loss of the proton gradient and cell membrane." However, there is no in depth discussion on different modes of action of the antibacterial properties. 

There are several paragraphs that contain speculative statements and missing citations.  Please provide citation on all the claims and arguments made. I.e., Line 281-284.

-Conclusion Line 381-397: Speculative statements. Please justify and explain in a scientific manner how the authors concluded that "By using protective cultures, the shelf life of products can be extended, thus avoiding food waste." and Line 388-390 how does the application of protective cultures could act to the elimination of refrigeration? Also, please identify the research gaps for the readers to explore potential areas in the field.
